# Repositioning of Ligands That Target the Spike Glycoprotein as Potential Drugs for SARS-CoV-2 in an In Silico Study

**DOI:** 10.3390/molecules25235615

**Published:** 2020-11-29

**Authors:** Gema Lizbeth Ramírez-Salinas, Marlet Martínez-Archundia, José Correa-Basurto, Jazmín García-Machorro

**Affiliations:** 1Laboratorio de Diseño y Desarrollo de Nuevos Fármacos e Innovación Biotécnológica, Escuela Superior de Medicina, Instituto Politécnico Nacional, Mexico City 11340, Mexico; gramirezs1204@egresado.ipn.mx (G.L.R.-S.); jcorreab@ipn.mx (J.C.-B.); 2Laboratorio de Medicina de Conservación, Escuela Superior de Medicina, Instituto Politécnico Nacional, Mexico City 11340, Mexico

**Keywords:** SARS-CoV-2, COVID-19, docking studies, FDA-approved drugs, spike glycoprotein

## Abstract

The worldwide health emergency of the SARS-CoV-2 pandemic and the absence of a specific treatment for this new coronavirus have led to the use of computational strategies (drug repositioning) to search for treatments. The aim of this work is to identify FDA (Food and Drug Administration)-approved drugs with the potential for binding to the spike structural glycoprotein at the hinge site, receptor binding motif (RBM), and fusion peptide (FP) using molecular docking simulations. Drugs that bind to amino acids are crucial for conformational changes, receptor recognition, and fusion of the viral membrane with the cell membrane. The results revealed some drugs that bind to hinge site amino acids (varenicline, or steroids such as betamethasone while other drugs bind to crucial amino acids in the RBM (naldemedine, atovaquone, cefotetan) or FP (azilsartan, maraviroc, and difluprednate); saquinavir binds both the RBM and the FP. Therefore, these drugs could inhibit spike glycoprotein and prevent viral entry as possible anti-COVID-19 drugs. Several drugs are in clinical studies; by focusing on other pharmacological agents (candesartan, atovaquone, losartan, maviroc and ritonavir) in this work we propose an additional target: the spike glycoprotein. These results can impact the proposed use of treatments that inhibit the first steps of the virus replication cycle.

## 1. Introduction

The COVID-19 disease is produced by the virus called SARS-COV-2, which emerged in China at the end of 2019. Currently, it has spread worldwide, so the World Health Organization (WHO) has declared a pandemic for which there is no treatment or vaccine [1].

The process to generate and test a vaccine will take at least a year [2]. However, by using computational tools such as repositioning, already known drugs (known information on pharmacokinetics, pharmacodynamics and toxicity) and those used for other pathologies can be proposed.

Repositioning drugs shows several advantages; for example, the time and production costs of the drugs are dismissed because the drugs are already available on the market [3]. In the case of aspirin, this drug was initially used as an analgesic and antipyretic and derived from repositioning strategies; currently, its new indication is as a treatment for colorectal cancer. Another example is Hydroxychloroquine an antiparasitic, which is now used in the treatment of antiarthritic systemic lupus erythematosus [4]. With regard to the treatment of COVID-19, different molecular targets have been proposed, such as modulation of immune defense, blocking viral cell entry, interfering with the endocytic pathway, targeting the cellular signaling pathway, blocking polyprotein posttranslational processing and various antiviral mechanisms [5]. SARS-CoV-2 infection can also affect the gastrointestinal tract, liver and pancreatic functions, leading to gastrointestinal symptoms and central and peripheral neurological manifestations, affecting the cardiovascular system, promoting renal dysfunction and, in general, resulting in a multitude of other clinical symptoms [6].

SARS-COV-2 belongs to the coronavirus family, and the genome is +ssRNA, nonsegmented, with a size of 27 to 32 kilobases [7]. The genome encodes four major structural proteins, spike (S), nucleocapsid (N), membrane (M) and envelope (E), which are required to make complete virus particles (see Figure 1A) [7,8]. The glycoprotein spike allows the penetration of host cells (obligatory step in the virus replication cycle), rendering it a powerful target for drug development.

The spike protein is comprised of the S1 and S2 subunits. The S1 subunit contains a signal peptide (SS), followed by an N-terminal domain (NTD) and receptor-binding domain (RBD). The RBD contains a core structure and a receptor-binding motif (RBM). The S2 subunit contains conserved fusion peptide (FP), heptad repeat 1 (HR1), a central helix (CH), a connector domain (CD), heptad repeat 2 (HR2), a transmembrane domain (TM), and a cytoplasmic tail (CT). At the boundary between S1 and S2 (S1/S2), there is a furin cleavage site at positions 681–684 (see Figure 1B) [9,10].

Conformationally, the spike glycoprotein is organized on the viral surface in homotrimers [11]. When the RBMs are hidden, the conformation is called down (receptor-inaccessible) (see Figure 1C). However, the homotrimer is asymmetric because it constantly undergoes structural rearrangement (up conformation) to fuse the viral membrane with the host cell membrane [12]. When two RBD domains are hidden (receptor-inaccessible), one RBD domain is exposed (receptor-accessible), resulting in the up conformation (see Figure 1D). This is because the RBD of S1 undergoes hinge-like movements [9]. In SARS-CoV, two hinge sites were characterized (hinge 1 site (354–361) and hinge 2 site (552–563)), which are responsible for the switch between the up and down conformations [13]. The RBD in the upper position recognizes the receptor through the receptor-binding motif (RBM), which binds to the outer surface of the claw-like structure of angiotensin-converting enzyme 2 (ACE2). The importance of RBM was demonstrated by comparing the sequences between SARS-CoV and SARS-CoV-2, and the five-amino-acid mutation (Leu455, Phe486, Gln493, Ser494 and Asn501) improves affinity for the receptor. Therefore, it is believed that SARS-CoV-2 is more infectious than SARS-CoV [10].

After recognition and binding of the RBM by ACE2, the spike glycoprotein needs to be proteolytically activated at the S1/S2 boundary such that S1 dissociates and S2 undergoes a dramatic structural change. These SARS-CoV-2 entry-activating proteases include cell surface type II transmembrane serine proteases (TMPRSS2) and lysosomal protease cathepsins. The structural change that S2 shows when dissociated is the exposure of the fusion peptide (FP) located at amino acid positions 816 to 855. Subsequently, the insertion of the FP into the host membrane is triggered, allowing the fusion of membranes (viral and cellular) to release the viral genome into the cytoplasm [14].

As mentioned above, the spike glycoprotein is in first contact with the host cell and is crucial for various processes, such as attachment, receptor binding, membrane fusion via conformational changes, internalization of the virus, and host tissue tropism [8,10]. Therefore, it is one of the main targets for the development of vaccines and antivirals.

In the search for potential compounds that could be used to treat COVID-19, in this work, we considered employing repositioning techniques, since they represent a new way of approaching drug compounds and targets that have been “derisked” during the development stages. In this work, we focused on drugs approved by the FDA that could target the spike glycoprotein during three specific events (conformational change from down to up, receptor recognition and membrane fusion).

## 2. Results

### 2.1. Molecular Modeling of Spike Glycoprotein

Figure 2 shows the conformational structure of the models obtained by molecular modeling. The 6VXX-fill model is seen in the down conformation, and the up conformation is shown for the 6VYB-fill model.

#### 2.1.1. Molecular Docking and Drug Selection

Once molecular docking was performed, the drugs with the highest affinity for the spike protein (mainly with amino acids belonging to the hinge site, the RBM and the FP) were selected. The drugs are listed in tables. The tables show the drug name, Zinc database identification number, use according to currently described pharmacological action, FDA approval number, inclusion in clinical studies related to COVID-19, affinity in kcal/mol (from highest to lowest affinity) and amino acid interactions from the spike glycoprotein. The general scheme shows the possible sites of interaction (Figure 1).

#### 2.1.2. Docking Simulations on the Hinge Site

Docking simulations were performed on hinge sites (1 and 2); however, only drug interactions were found at hinge site 1 (see Table 1 and Table 2). The affinity binding scores and molecular interactions of the drugs for the down conformation are detailed in Table 1.

Figure 3A shows varenicline, which binds to chains B and C in hinge site 1, RBM and NTD with an affinity of −7.6 kcal/mol in the down conformation via Van der Waals forces (Pro230C, Lys458B and Arg466B), π-interactions (Ile231C and Tyr200C), alkyl interactions (Trp353B and Arg355B), π-cation interactions (Asp467B and Arg355B) and finally hydrogen bonds (Gly232C, Glu465B, Asp198C and Gly199C). In addition, Figure 3B shows that docosahexaenoic acid interacts by Van der Waals forces with the amino acids Pro230B, Val227B, His207B, Asp228B, Leu229B, Phe168B, Tyr170B, Ile128B, Ile203B, Phe194B, Ile119B, Phe192B, Trp104B and Asn121B by electrostatic interactions and hydrogen bonds with Arg357B and finally alkyl interactions with Leu179B and Val126B. Another interesting drug was sulbactam (Figure 3C), which interacts with in the hinge 1 site, the RBM and the NTD by Van der Waals forces with the amino acids Asp198C, Pro426B, Leu518B, Glu516B, Phe515B, Tyr396B and Ser514B and two hydrogen bonds with the amino acids Arg466B and Arg355B.

Table 2 shows compounds that interact mainly with amino acids belonging to hinge site 1, the RBM site and the NTD of the up conformation. Figure 4A and Table 2 show that zafirlukast binds to chain A and the hinge 1 site (−10.3 kcal/mol) in the up conformation. zafirlukast undergoes Van der Waals interactions (Phe168B, Arg357A, Ile128B, Leu176B, Ile119B, Asn121B, Arg102B, Phe175B, Phe192B, Ser172B and Gln173B), hydrogen bonds (Pro174B, Asp228B and Val227B), π-π interactions (Tyr170B), π-sigma interactions (Leu226B) and alkyl interactions (Pro174B, Leu226B, Val126B, Met177B and Pro230B). Tigecycline (Figure 4B) joins the A-chain at the hinge 1 site with an affinity of −9.7 kcal/mol in the up conformation. The drug interacts through Van der Waals forces with the amino acids Glu516A, Tyr200B, Arg357A, Pro426A and Gly232B, hydrogen bonds with Pro230B, Gly199B, Ile231B, Phe429A, Phe515A, Thr430A, Phe464A and Asp198B, and π-π interactions with Tyr396A and π-Cation with Arg355A.

#### 2.1.3. Docking Simulations on the RBM

Table 3 and Table 4 show the drugs selected for the RBM in the down and up conformations, respectively.

The drug with the highest binding score for the RBM site in down conformation was naldemedine (Figure 5A), which interacts by Van der Waals forces with Gln493C, NAG1303A, Leu492C, Gly476C, Ala475C, Tyr473C, Asp467C, Asp420C, Glu465C, Tyr453C, Phe456C and Arg457C; forms hydrogen bonds with Arg454C, Leu455C and Tyr421C; undergoes alkyl interactions with Pro491C and Lys458C; and undergoes π-cation interactions with Lys417C and Arg454C. The second drug that showed the highest affinity was conivaptan (Figure 5B), which interacts with chain C in the RBM region by Van der Waals forces (Pro491C, Gln493C, Tyr453C, Lys458C, Ile472C, Asn422C and Asp420C), hydrogen bonds (Phe456C, Arg457C and Tyr421C) and π interactions (Asp467C, Leu455C, Lys417C and Tyr421C).

Regarding the up conformation, Figure 6A shows that tedizolid phosphate binds to chain B in the RBM and displays an affinity of −9.7 kcal/mol. The drug interacts through Van der Waals forces (Asp405, Gly416B, Ile402B, Ile418B, Gln493B, Ser494B, Tyr495B, Leu452B, Tyr451B and Tyr453B) and π-cation interactions (Arg408B and Lys417B) and forms hydrogen bonds with Gln409B, Gln414B, Lys417B and Arg403B. The drug atovaquone (Figure 6B) binds to the B chain in the RBM by interacting through Van der Waals forces (Leu455B, Ser494B, Ala352B, Asp467B, Phe456B, Asn450B, Pro491B, Thr478B, Gly476B and Ser477B), π-π interactions (Tyr351B), hydrophobic interactions (Leu492B and Leu452B) and hydrogen bonds (Arg454B and Pro479). This drug showed the second highest binding affinity, as indicated in Table 4.

Regarding the drug cefotetan, although it did not show one of the highest affinities, it interacts with the most crucial amino acids for the formation of the spike-ACE2 complex. Cefotetan (Figure 6C) is bound in the up conformation by Van der Waals forces (Ser349B, Tyr489B, Leu455B, Pro491B, Gly476B, Ser477B, Phe490B, Gln493B, Leu492B, Tyr351B, Tyr449B and Asp467B) and forms hydrogen bonds (Tyr449B, Asn450B, Arg454B and Thr478B), alkyl interactions (Pro479B and Leu452B) and π-sulfur interactions (Phe456B and Tyr449B).

#### 2.1.4. Docking Simulations on the FP

The drugs that displayed the highest affinity for the fusion peptide in both the down and up conformations are summarized in Table 5 and Table 6, respectively. The results from the docking analyses are ordered from highest to lowest affinity. Some representative drug interactions are described below. The drug saquinavir (Table 5 and Figure 7A) shows the highest affinity to the FP site (−11.1 kcal/mol), binding to the amino acids Cys851A and Leu849A in the down conformation. Saquinavir interacts through Van der Waals forces (Lys835A, Cys851A, Ala852A, Leu828A, Gly832A, Ile834A, Pro862A, Pro863A, Lys854A, Arg646C, Ile850A, Glu619C, Ser591C and NAG1309C) and forms hydrogen bonds (Asp843A, Tyr837A, Asp614C, Asn616C, Val615C, Gln644C, Gly648C and Thr645C), π-alkyl interactions (Val860A and Tyr837A), π-sigma interactions (Leu849A) and π-cation interactions (Tyr837A). The drug difluprednate showed less affinity for the same site (−8.9 Kcal/mol) (Table 5 and Figure 7B) and in the down conformation, forming Van der Waals interactions with amino acids Ala829A, Leu849A, Lys835A, Ala852A, Gly832A, Ile834A, Val860A, Asp614C, Ala668C, Thr866A, Ala647C and Cys840C, hydrogen bonds with amino acids Phe833A and Arg646C, halogen interactions with Ile850A and finally π-alkyl interactions with Cys851A, Leu828A and Tyr837A.

Additionally, saquinavir (Table 6 and Figure 8A) showed the highest affinity in the up conformation. Saquinavir interacts by Van der Waals forces (Asn856C, Asn960C, Gln853C, Leu858C, Ala852C, Thr859C, Asp614B Leu959C, Thr732C, Phe833C, Val952C, Asn955C, Ile834C, Gly832C, Ala831C, Arg646B, Leu849C, Gln836C, Ala570B and Tyr837C), forming hydrogen bonds with the following residues: Arg847C, Asp848C anThr732C; alkyl interactions (Lys854C, Val860C, Ala956C, Lys835C and Cys851C); and π-sulfur interactions (Cys851C and Cys840C). Another drug that reached the up conformation was maraviroc (Figure 8B), which interacts with the amino acids Val952C, Asn955C, Leu959C, Thr732C, Leu858C, Ala570B, Asn856C, Val860C, Asp848C, Arg847C, Tyr837C and Gln836C through Van der Waals forces and forms hydrogen bonds with the amino acids Arg646B and Gln836C and alkyl interactions with the amino acids Lys835C, Ala956C, Phe833C, Val963C, Ala852C, Cys851C, Leu849C and Cys840C.

## 3. Discussion

Spike glycoprotein is the structural protein of the SARS-CoV-2 virus that allows adhesion and binding to the receptor, making it crucial in the first step of infection [15]. Currently it is considered one of the main targets for vaccine design. However, this protein evades the immune response due to its dynamic structure, for example, when it is found in the down conformation (receptor-inaccessible) where it hides RBD (hidden RBD) and prevents the binding of neutralizing antibodies. Additionally, when the spike glycoprotein is found in the up conformation (accessible receptor), the RBD has high affinity for the receptor (higher affinity than SARS-CoV). In addition, the virus can enter through two routes, receptor-mediated endocytosis and direct fusion to the cell membrane, because the fusion peptide can be exposed by cell surface proteases (depending on the cell type). Therefore, vaccines targeting the membrane fusion S2 subunit can be developed; however, the S2 subunit is less immunogenic than the S1 subunit [16,17].

In this work, we employed an in silico strategy called drug repositioning, in which we searched for available drugs and targets of the spike glycoprotein (hinge sites, RBM and FP of interest). From this analysis, it was possible to identify 43 drugs, of which 11 are included in reported clinical trials [18].

Regarding the hinge region in the down conformation, 3 drugs were found to undergo possible binding and prevent the change to the up conformation, therefore avoiding exposure of the receptor binding site. The drug with the highest affinity is varenicline (−7.6 kcal/mol); however, it is not found in any clinical study, unlike docosahexaenoic acid (DHA) and sulbactam, which are being studied as complementary components of primary treatments. In the DHA clinical trial, the main treatment was fenretinide (LAU-7b), a synthetic retinoid derivative that has been used in the treatment of some types of cancer and cystic fibrosis. DHA is an omega-3 fatty acid and can be included in the diet. Another fatty acid that has been reported in six clinical trials is icosapent ethyl (Vascepa™), although not as part of a dietary food supplement. Its effect is observed on inflammatory biomarkers in individuals with COVID-19. Arachidonic acid (AA) and other unsaturated fatty acids (especially eicosapentaenoic acid, EPA and DHA) are known to inactivate enveloped viruses and inhibit proliferation, so they can serve as endogenous antivirals [19]. Furthermore, sulbactam is a semisynthetic beta lactamase inhibitor that has been commonly used in conjunction with intravenous ampicillin [20] to avoid secondary infection in patients with cytokine adsorption in severe COVID-19 pneumonia requiring extracorporeal membrane oxygenation (CYCOV).

As mentioned before, the spike glycoprotein has been shown to be very dynamic due to its hinge sites, and preventing the switch from up to down conformation can help avoid hide the receptor binding site, while combination therapy (with neutralizing antibodies) can prevent viral attachment. There were 5 drugs that were found to bind best at the hinge site in the upper position (zafirlukast, tigecycline, betamethasone, triamcinolone acetonide, atazanavir), three of which are in clinical trials (betamethasone and triamcinolone acetonide, both corticosteroids; and atazanavir, an antiviral). Betamethasone is a steroid from a group of corticosteroids that is used in pregnancies complicated by SARS-CoV-2 due to its immunosuppressive and anti-inflammatory properties. However, a potential diabetogenic effect was found, in addition to the glycemic effects of SARS-CoV-2 and other coronaviruses [21].

Therefore, the use of betamethasone in clinical trials is not recommended and has not been reported. In contrast, the short-term use of dexamethasone in severe, intubated COVID-19 patients has recently been recommended due to the anti-inflammatory effect that would limit the production and damaging effect of cytokines, but dexamethasone will also inhibit the protective function of T cells and block B cells from making antibodies [22]. The use of betamethasone and dexamethasone, given the recognition of hinge site blockage, has not been explored, and the two corticosteroids have the same molecular weight and high structural similarity (the only difference is the orientation of the methyl groups on carbon 16), which could have similar effects. The use of inhaled corticosteroids has been associated with lower expression of ACE2 and TMPRSS2, but treatment with triamcinolone acetonide did not decrease the expression of either gene [23]. On the other hand, the use of atazanavir (protease inhibitor) has been reported in computer studies of inhibitory potency activity with a Kd of 94.94 nM against the SARS-CoV-2 3C-like protease inhibitor [24]. Clinical trials were beginning recruiting in October.

On the other hand, we found 10 better evaluated drugs from docking studies at the RBM site in the down conformation, which, although it would not prevent the change to the up conformation, would prevent receptor binding. Nine of the drugs show interaction with two amino acids (Leu455 and Gln493) out of the five reported (Leu455, Phe486, Gln493, Ser494 and Asn501) as crucial for the RBM site in the formation of a stable complex between spike viral glycoprotein and human ACE2 [10,25]. Only saquinavir interacts with the key amino acid Leu455. Despite these interactions, the affinity for the glycoprotein is different, with the highest affinity being (−12.7 kcal/mol) and the lowest affinity being suvorexant, riciguan, glibenclamide and candesartan (−10.6 kcal/mol). Candesartan was included in four clinical studies due to the selective antagonist function of angiotensin II AT1 receptors. In another study, pancuronium bromide was included as a muscle relaxant in pregnant and postpartum women hospitalized with flu syndrome and COVID-19 (clinical trial NCT04462367). Interestingly, there are some scientific reports available about its possible potential as a drug to treat COVID-19 [26].

When the structural conformation of the spike glycoprotein in the S1 domain changes to the upper position, it shows high affinity for the receptor, which can be blocked by drugs that bind to the RBM in up position. Six drugs were found to show affinity for the RBM site in the up position. However, the affinity values are lower than the affinity shown in the down position (−9.7 kcal/mol in the up position vs. −12.7 kcal/mol in the down position). The drugs with the highest affinity were tedizolid phosphate and atovaquone (−9.7 kcal/mol). Atovaquone is in a clinical trial (NCT04339426, in combination with azithromycin) to treat COVID-19. Losartan has been reported in 16 clinical trials and has been tested for the treatment of SARS-CoV-2 pneumonia in noncritically ill subjects. On the other hand, the drug with the lowest affinity was rosiglitazone (−8.3 kcal/mol), which was used to regulate glucose in type 2 diabetes; however, its use is discontinued [27].

Considering another point in the viral replication cycle, we looked for drugs capable of binding to FP, taking into account that it is a conserved sequence, and SARS-CoV shows two FP sites. FP1 (amino acids 798 to 818) and FP2 (amino acids 816 to 835) have the structural characteristics of an active fusion region; furthermore, they postulate that the regions function cooperatively as an extended FP (FP1-2) [28,29]. In the case of the SARS-CoV-2 spike glycoprotein, Tang and collaborators recently determined that FP is found in the region located in amino acids 816 to 855, which corresponds to the region 798–835 for the SARS-CoV spike glycoprotein. The amino acids Leu803, Leu804, Phe823, Cys822, Cys833, Asp830 and Leu831 in SARS-CoV (corresponding to the amino acids Leu821, Leu822, Phe823, Cys840, Asp848, Leu849, and Cys851 in SARS-CoV-2) have been shown to be essential for membrane fusion processes through biophysical studies and site directed mutagenesis studies. [28]. In addition, these amino acids are conserved in SARS-CoV, MERS-CoV and SARS-CoV-2 [14].

Notably, nine drugs were bound in the down conformation within the FP and interact mainly with two highly conserved amino acids (Cys851A and Leu849A), which are crucial for the process of membrane fusion in the family of coronaviruses [14]. However, only nebivolol is currently in an observational clinical trial to compare SARS-CoV-2-positive outpatients and compare all-cause hospitalization and mortality rates between doses of ACEI/ARB- vs. non-ACEI/ARB-based regimens. In this computational study, the drug with the highest affinity was saquinavir (−11.1 kcal/mol), and the lowest affinity was difluprednate (−8.9 kcal/mol). However difluprednate interacts with an additional amino acid to modulate the membrane fusion process (Cys840A) among a total of three critical amino acids (Cys840A, Cys851A and Leu849A). To be able to use these drugs in future clinical studies (in addition to the interactions found), commercial presentation must be considered. Such is the case for terconazole, which is administered as a cream and suppository vaginal presentation (https://www.accessdata.fda.gov), so it cannot be used for the treatment of COVID-19 with that presentation.

On the other hand, when the protein is in the up position, 10 drugs were found to bind FP, of which five (maraviroc, ritonavir, bosentan, fosinopril sodium, ceftazidime) are in clinical trials. Maraviroc, a C-C chemokine receptor 5 (CCR5) antagonist, is well tolerated without significant side effects in its current use in patients with HIV; in patients with COVID-19, it has been used as a drug against the main protease (Mpro) of SARS-CoV-2 (determined by computer methods) [30]. The drug found in the largest number of clinical trials is ritonavir, which is mostly used in combination with lopinavir. Both are protease inhibitors to treat HIV and have been used against other coronaviruses (SARS-CoV and MERS-CoV) [31]. Use against SARS-CoV-2 also seeks to inhibit cellular proteases to prevent completion of the viral replication cycle; however, some clinical trials have not been concluded, and others have concluded that there is no difference in using ritonavir in hospitalized adult patients with severe COVID-19 and standard medical care treatment [32]. Another class of drugs that have been used for the treatment of COVID-19 is ACE inhibitors, such as fosinopril sodium, which is being studied in two clinical trials (18) as a means to block the virus receptor. Therefore, these three drugs, in addition to the effects described in the clinical trial, could prevent fusion of the viral membrane with the cell membrane.

Furthermore, the drug bosentan is a dual endothelin receptor antagonist used in the treatment of pulmonary hypertension (PHT) [18], and ceftazidime is an antibiotic used to treat lower respiratory tract infections; both of these drugs and others (i.e., budesonide, cefdinir, cefepime, clindamycin, clobazam, dexamethasone, dexmedetomidine, fosfomycin, dextroamphetamine, etc.) are part of a clinical study called Pharmacokinetics, Pharmacodynamics, and Safety Profile of Understudied Drugs Administered to Children Per Standard of Care (POPS or POP02) in the treatment of COVID-19.

Interestingly, saquinavir has been found to have the highest affinity for FP in both conformations (up and down), showing an affinity of −11 kcal/mol, and it binds to four crucial amino acids in the membrane fusion process (Cys851C, Cys840C, Leu849C and Asp848C). Furthermore, it can bind to the RBM in the down position. Therefore, this drug is a potential candidate to inhibit the FP and RBM sites of the spike glycoprotein.

Considering the data reported, the use of multitarget drugs can be proposed, and by interfering at different points in the replication cycle, it could exert a better effect. It is even possible to propose combined therapies that prevent both viral entry (spike target) and replication cycle events inside the host cell as preventive and therapeutic treatments.

## 4. Materials and Methods

### 4.1. Molecular Modeling of Spike Glycoprotein SARS-CoV-2

Two crystal structures that represent down and up conformational states of the spike glycoprotein were used. ID PDB: 6VYB represents the up conformation, whereas ID PDB: 6VXX constitutes the down conformation. Both crystals showed missing fragments and two disulfide bridges (480–488 and 840–851), which are important in the RBM and FP regions, respectively; thus, these were completed using Program Modeller 9.23 (scripts: loop.py, segment.py and model-disulfide.py) [33]. After adding the missing residues and two disulfide bridges, the resultant tridimensional (3D) models were denominated as 6VXX-fill and 6VYB-fill down and up conformations, respectively. In model 6VYB, which fills chain B, the RBD site is exposed. The glycosylated crystals were preserved and added by the Chimera program [34] to models 6VXX-fill and 6VYB-fill.

### 4.2. Selection of Drugs to Be Repositioned

The 3D structures of the drugs were obtained from the database Zinc [35] accessed March 2020, from which the subset fda was downloaded, which includes drugs that are approved by the FDA, per DrugBank; we then selected those that comply with the rules of Lipinski and have suitable Absorption, Distribution, Metabolism, Elimination and Toxicity (ADMET) properties of binding to albumin, and solubility in water and in DMSO [36]. Additionally, the subset fda was submitted to the Data Warrior program to predict drugs that had mutagenic, tumorigenic, reproductive or irritant effects. The drugs that presented those effects were eliminated, and for docking studies, only those drugs that were safe were docked. In total, 1372 drugs were analyzed; according to the criteria, 1322 drugs were selected to be coupled to the spike glycoprotein by docking studies.

### 4.3. Molecular Docking

Molecular docking studies were carried out using the AutoDock Vina program [37].The proteins were prepared with Autodock Tools 1.5.2, polar hydrogens were included, and Kollman charges were calculated. The free energy values of docking were calculated using a scoring function implemented in Autodock. The drugs were added to the hydrogens using the OpenBabel program [38] and then the pdbqt files were prepared using the prepare_ligand4.py script [37]. Finally, the conformations for each ligand on the target proteins were analyzed.

The spike glycoprotein was rigid, and the ligands were flexible. The drugs with the highest affinity and those that reach key amino acids that influence spike glycoprotein function were selected. The docking simulation focused on the hinge sites, RBM and FP for the 3 chains of the spike glycoprotein trimer in the up and down conformations. In total, 1322 dockings were achieved for the A, B and C chains for the down and up conformations. The docking was conducted with exhaustiveness of 28, number modes of 20 and energy range of 6.

### 4.4. Drug Selection

After completing the docking simulations, Discovery Studio Visualizer [39] and Chimera [34] programs were used to visualize the docking results. Drugs with the highest affinity (more negative) and those interacting with key amino acids to function at the sites of interest (hinge, RBM and FP) were selected.

## 5. Conclusions

By using in silico tools, a subgroup of drugs previously approved by the FDA for use in various human diseases has been analyzed. From this group of drugs, we were able to identify potential drugs that could inhibit the function of spike glycoprotein interference and the entry of SARS-CoV-2 into host cells to inhibit viral infection. Among these potential drugs, some are reported in clinical trials as therapies in conjunction with other drugs (DHA, sulbactam and atovaquone); others are part of a primary therapy due to their antihypertensive effects (fosinopril sodium and candesartan), anti-inflammatory effects (betamethasone, triamcinolone acetonide), and effects on proteases (atazanavir and ritonavir). Additionally, maraviroc is used as a CCR5 antagonist and was determined by computer methods to be a possible protease inhibitor. On the other hand, even though saquinavir has not been reported in clinical trials, it has been suggested from theoretical and in silico studies to exert a 3C-like protease effect. Therefore, in this work, we propose other therapeutic targets that could bind the spike protein and that may have a combined effect with the other reported targets/uses.

## Figures and Tables

**Figure 1 molecules-25-05615-f001:**
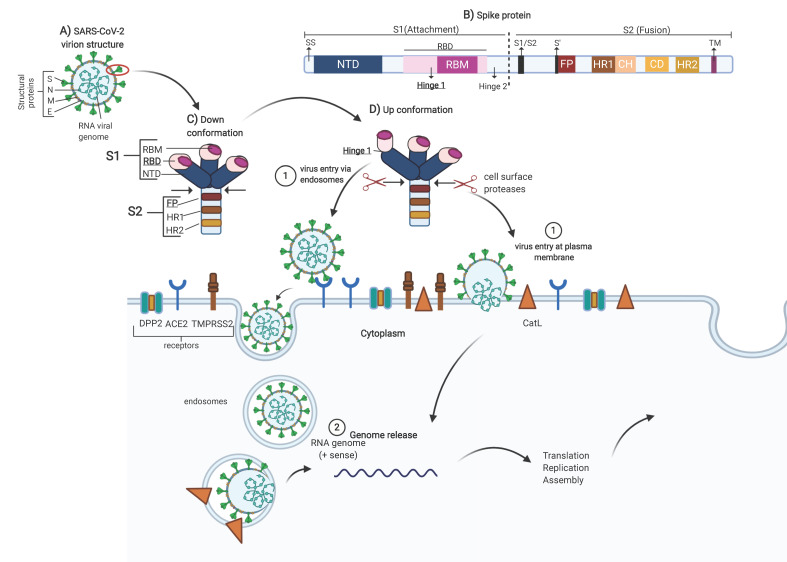
Structural spike glycoprotein. (**A**) Structural proteins, spike (S), nucleocapsid (N), membrane (M), envelope (E). (**B**) Spike protein sequence; S1 subunit contains: signal sequence (SS), N-terminal domain (NTD), receptor binding domain (RBD), receptor-binding motif (RBM); S2 subunit contains: S1/S2 cleavage site, fusion peptide (FP), heptad repeat 1 (HR1), central helix (CH), connector domain (CD), heptad repeat 2 (HR2), transmembrane domain (TM). (**C**) In the down conformation, the homotrimer hidden the RBD (receptor-inaccessible). (**D**) In the up conformation, the homotrimer is asymmetric, two RBD domains are hidden (receptor-inaccessible), and one RBD domain is exposed (receptor-accessible). The sites of interest for this work are underlined and in bold (figure created with BioRender.com).

**Figure 2 molecules-25-05615-f002:**
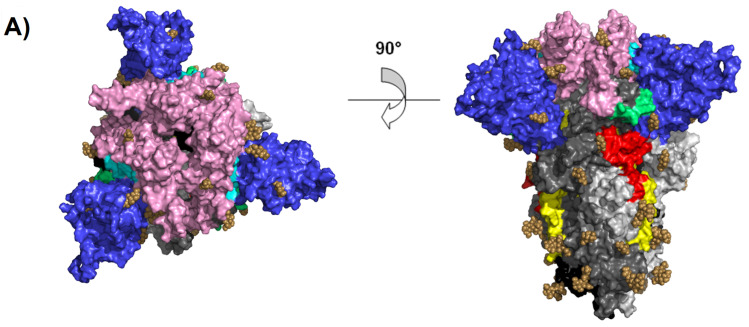
Spike glycoprotein trimer top (left) and side (right) views. (**A**) The trimer is observed in the down conformation (6VXX-fill). (**B**) The trimer is shown in the up conformation (6VYB-fill). The region of the ectodomain was modeled, and the following regions were observed: RBD (pink), NTD (blue), HR1 (yellow), FP (red), hinge 1 (cyan) and hinge 2 (lime). The figure is shown in black and white colors for a better view of the chains. Chain A is colored black, chain B is colored dark gray, and chain C is colored light gray.

**Figure 3 molecules-25-05615-f003:**
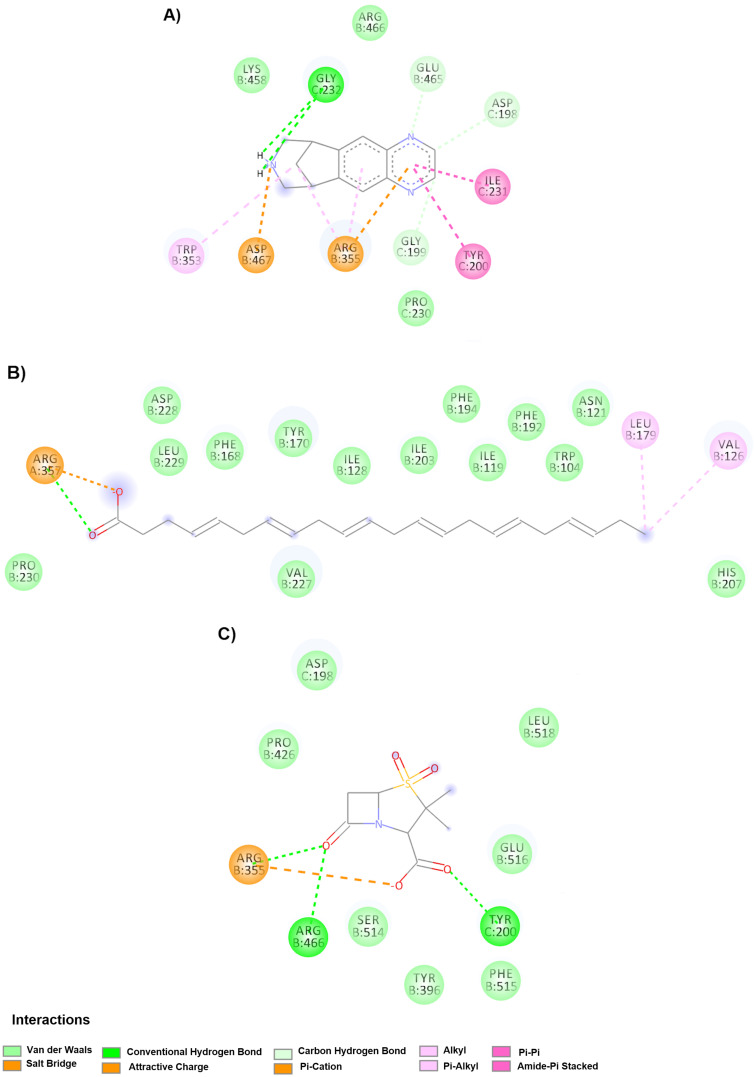
Molecular interactions of the drugs in the hinge 1 site with spike glycoprotein in the down conformation. (**A**) Interactions of the varenicline-spike glycoprotein complex are observed. (**B**) Binding mode of the docosahexaenoic acid-spike glycoprotein complex. (**C**) Molecular interactions with the drug sulbactam and spike glycoprotein. The best compounds directed to the hinge site bind preferentially to hinge site 1 because a cavity is formed, composed of hinge site 1, RBM, and NTD, and the drugs studied bind with greater affinity to this cavity.

**Figure 4 molecules-25-05615-f004:**
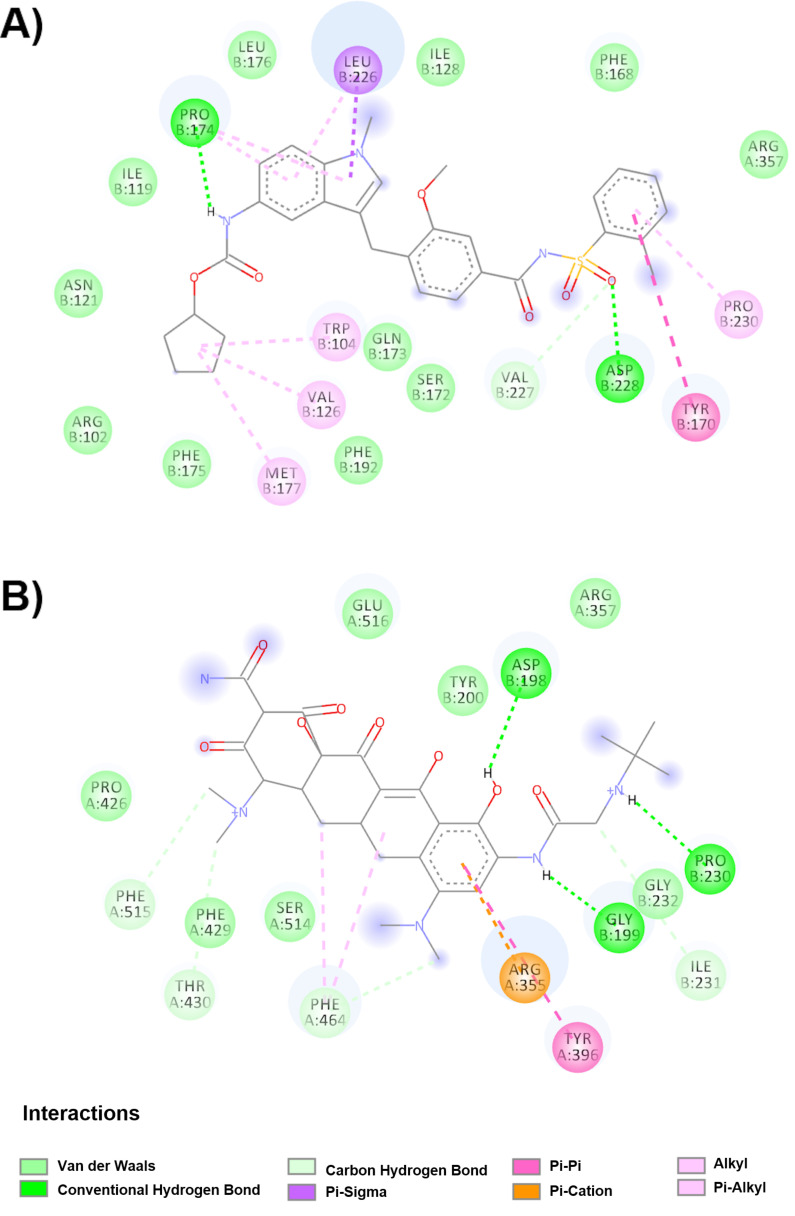
Molecular interactions of the drugs with the hinge 1 site spike glycoprotein in the up conformation. (**A**) Interactions of the zafirlukast spike glycoprotein complex are observed. (**B**) Molecular interactions with the drug tigecycline and in the spike glycoprotein.

**Figure 5 molecules-25-05615-f005:**
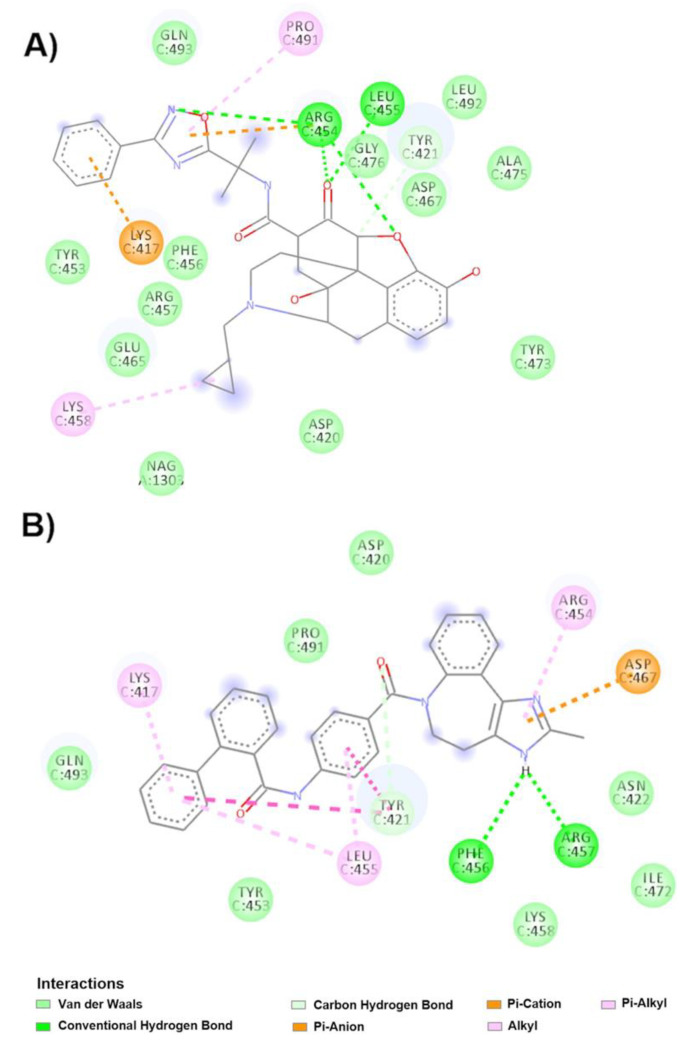
Molecular interactions of the drugs with RBM in the spike glycoprotein in the down conformation. (**A**) Interactions of naldemedine with the spike glycoprotein complex are observed. (**B**) Molecular interactions with conivaptan and spike glycoprotein chain C.

**Figure 6 molecules-25-05615-f006:**
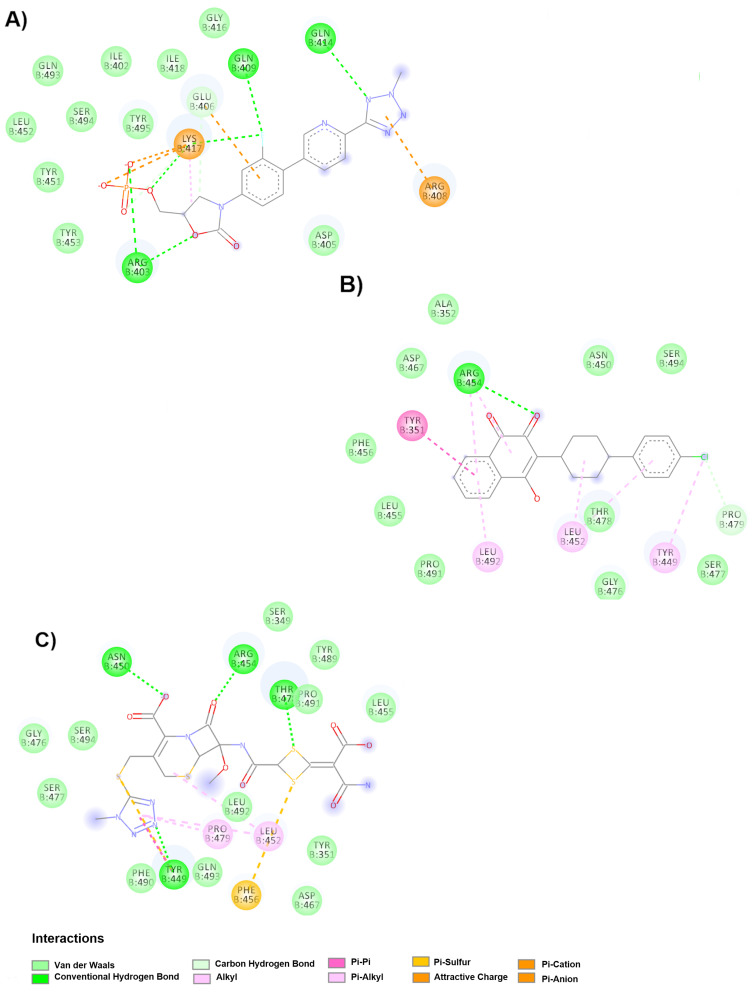
Molecular interactions of the drugs with RBM in spike glycoprotein in the up conformation. (**A**) Interactions of the tedizolid phosphate and spike glycoprotein complex are observed. (**B**) Interactions of the atovaquone-spike glycoprotein complex are shown. (**C**) Interaction of cefotetan with the spike glycoprotein.

**Figure 7 molecules-25-05615-f007:**
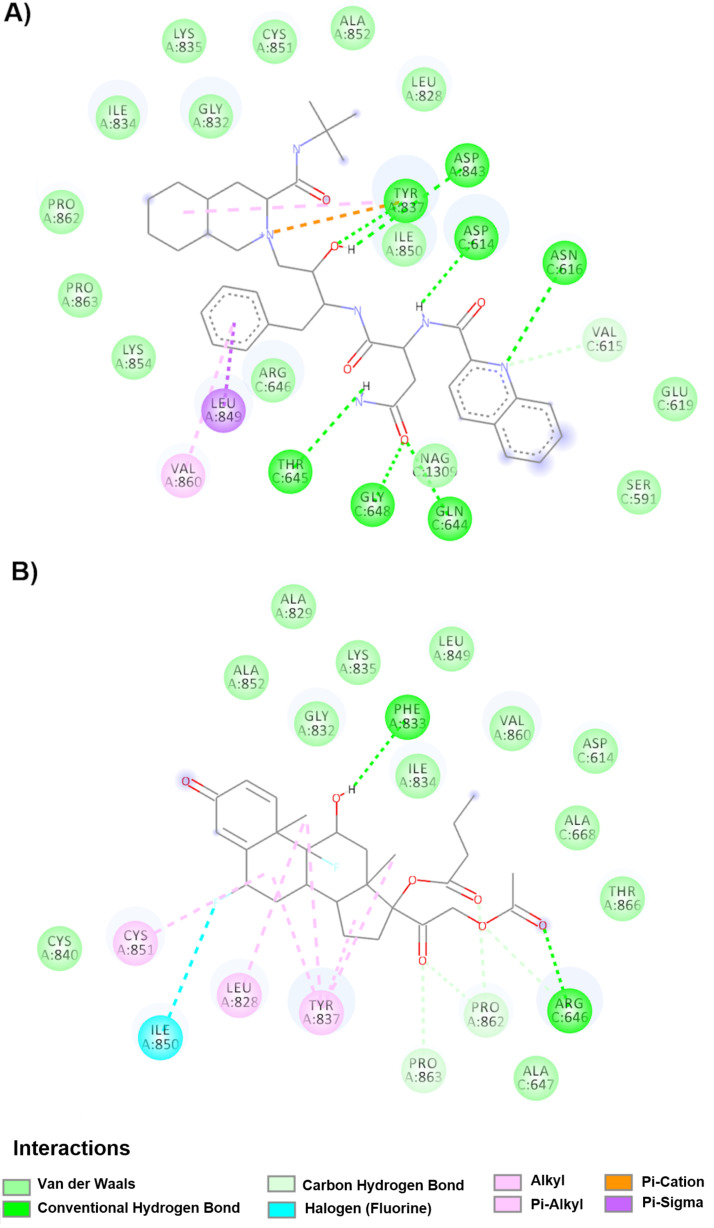
Molecular interactions of drugs with the spike glycoprotein in FP (down conformation). (**A**) Interactions of the saquinavir-spike glycoprotein complex are observed. (**B**) Molecular interactions with the drug difluprednate and chain A of the spike glycoprotein.

**Figure 8 molecules-25-05615-f008:**
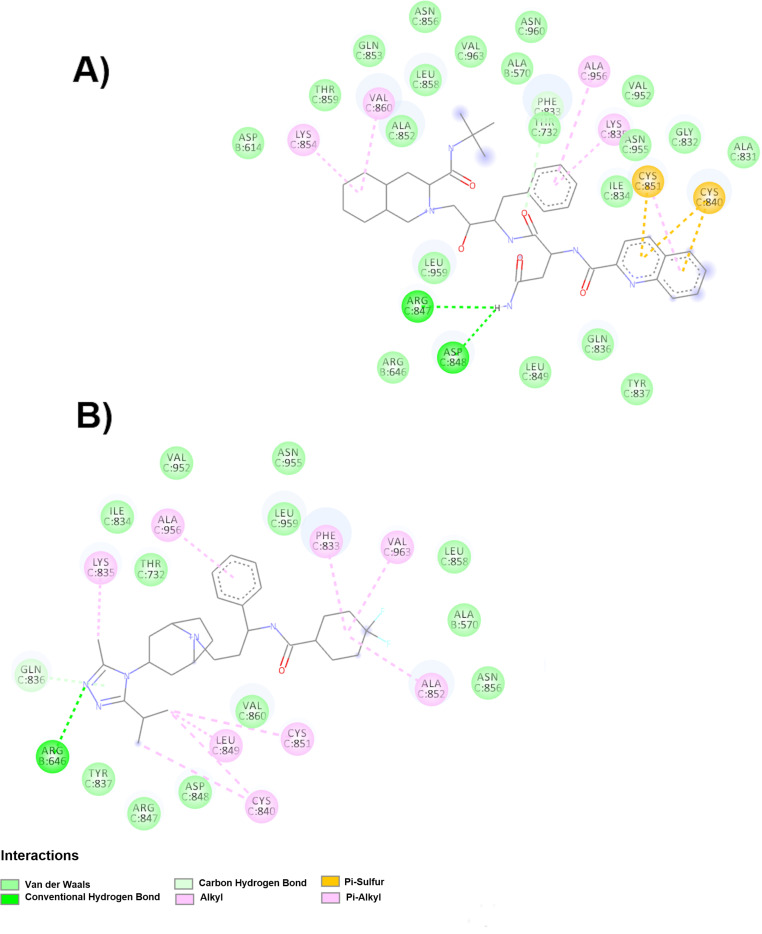
Molecular interactions of the drugs with spike glycoprotein in the FP (up conformation). (**A**) Interactions of the saquinavir-spike glycoprotein complex are observed. (**B**) Molecular interactions with the drug maraviroc with the spike glycoprotein.

**Table 1 molecules-25-05615-t001:** Selected drugs for the hinge site in the down conformation.

Drug	ID_ZINC ^a^	Pharmacological Action	FDA-Approved Drugs ^b^	Clinical Studies Related to COVID-19 ^c^	Affinity (kcal/mol)	Amino Acid Interactions with Spike Glycoprotein
Varenicline	ZINC1481833	Agonist for α4,β2 receptors nicotinic and receptor agonist	NDA #021928	NF	−7.6	Lys458B, Arg355B, Pro230C, Arg466B, Ile231C, Tyr200C, Trp353B, Asp467B Gly232C, Glu465B, Asp198C, Gly199C
Docosahexaenoic acid (DHA)	ZINC4474564	Omega-3 essential fatty acid	NF	NCT04417257 NCT04495816	−7.2	Arg357A, Pro230B, Val227B, His207B, Asp228B, Leu229B, Phe168B, Tyr170B, Ile128B, Ile203B, Phe194B, Ile119B, Phe192, Trp104B, Asn121B, Leu179B, Val126B
Sulbactam	ZINC897244	β-Lactamase inhibitor	ANDA #065074	NCT04324528	−7	Arg355B, Asp198C, Pro426B, Leu518B, Glu516B, Phe515B, Tyr396B, Ser514B, Arg466B

^a^ ID_ZINC (http://zinc.docking.org/), accessed March 2020. ^b^ Information retrieved from Drugs@FDA (https://www.accessdata.fda.gov), accessed September 2020. Latest New Drug Application (NDA) and Abbreviated New Drug Application (ANDA). ^c^ Information retrieved from https://clinicaltrials.gov/, accessed September 2020.

**Table 2 molecules-25-05615-t002:** Selected drugs for the hinge site in the up conformation.

Drug	ID_ZINC ^a^	Pharmacological Action	FDA-Approved Drugs ^b^	Clinical Studies Related to COVID-19 ^c^	Affinity (kcal/mol)	Amino Acid Interactions with Spike Glycoprotein ^d^	
Zafirlukast	ZINC896717	Leukotriene Receptor Antagonists	NDA 020547	NF	−10.3	Arg357A, Phe168B, Ile128B, Leu176B, Ile119B, Asn121B, Arg102B, Phe175B, Phe192B, Ser172B, Gln173B, Pro174B, Asp228B, Val227B, Tyr170B, Leu226B, Pro174B, Val126B, Met177B, Pro230B	
Tigecycline *	ZINC14879972	Blocking entry of amino-acyl transfer RNA	ANDA #091620	NF	−9.7	Arg357A, Arg355A, Glu516A, Tyr200B, Pro426A, Gly232B, Pro230B, Gly199B, Ile231B, Phe429A, Phe515A, Thr430A, Phe464A, Asp198B, Tyr396A	
Betamethasone	ZINC3876136	Secretory leukocyte protease inhibitor	NDA 020934	NCT04462367	−9.1	Arg355A, Ser514A, Tyr200B, Gly199B, Asp198B, Asp428A, Glu516A, Phe529A, Leu517B, Phe515A, Thr430A, Tyr396A, Pro426A, Phe464A	
Triamcinolone Acetonide	ZINC3882036	Activates the glucocorticoid receptor	NDA #012104	NCT04395482 NCT04528368	−8.9	Arg355A, Gly199B, Asp198B, Phe464A, Pro426A, Phe429A, Ser514A, Glu516A, Asp428, Thr430A, Phe515A, Tyr396A	
Atazanavir	ZINC3941496	Protease inhibitors	ANDA #091673	NCT04459286 NCT04452565 NCT04468087	−8.9	Phe562A, Pro225B, Phe562A, Pro521A, Ile128B, Ile203B, Ile119B, Asn121B, Arg102B, Trp104B, Phe175B, Phe192B, Pro174B, Gln173B, Ser172B, Tyr170B, Lys41B, Val227B, Asp228B Leu226B, Val126B, Met177B	

* Only powder for intravenous route administration. ^a^ ID_ZINC (http://zinc.docking.org/), accessed March 2020. ^b^ Information retrieved from Drugs@FDA (https://www.accessdata.fda.gov), accessed September 2020. Latest New Drug Application (NDA) and Abbreviated New Drug Application (ANDA). ^c^ Information retrieved from https://clinicaltrials.gov/, accessed September 2020. ^d^ Important amino acids involved for the formation of the spike glycoprotein-ACE2 complex.

**Table 3 molecules-25-05615-t003:** Selected drugs for the RBM in the down conformation.

Drug	ID_ZINC ^a^	Pharmacological Action	FDA-Approved Drugs ^b^	Clinical Studies Related to COVID-19 ^c^	Affinity (kcal/mol)	Interactions with Spike Glycoprotein ^d^	
Naldemedine	ZINC100378061	Opioid receptor antagonist	NDA 208854	NF	−12.7	**Gln493C**, Leu455C, Leu492C, Gly476C, Ala475C, Tyr473C, Asp467C, Asp420C, Glu465C, Tyr453C, Phe456C, Arg457C, Arg454C, Tyr421C, Pro491C, Lys458C, Lys417C, NAG1303A	
Conivaptan	ZINC12503187	Vasopressin receptor antagonist	NDA #022016	NF	−12.1	**Leu455C, Gln493C**, Asp420C, Pro491C, Arg454C, Asp467C, Lys417C, Tyr421C, Tyr453C, Phe456C, Phe457C, Asn422C, Ile472C, Lys458C	
Tipranavir	ZINC100022637	Protease inhibitor	NDA #021814	NF	−10.9	**Leu455C, Gln493C**, Tyr453C, Asp420C, Try421C, Ser477C, Asn422C, Gly476C; Ala475C, Pro491C, Lys417C, Tyr473C, Arg454C, Phe456C, Arg457C, Asp467C	
Vorapaxar	ZINC3925861	Thrombin Receptor Antagonist	NDA 204886	NF	−10.9	**Leu455C, Gln493C**, Glu406C, Tyr453C, Tyr495C, Arg403C, Ala372A, Phe456C, Lys458C, Ile468C, Gly476C, Ser477C, Thr478C, Glu465C, Pro491C, Tyr421C, Lys417C, Arg454C, Asp467C, Arg457C	
Pancuronium bromide	ZINC4097383	Muscle relaxant	ANDA #072320	NCT04462367	−10.8	**Leu455C, Gln493C**, Asp420C, Phe456C, Ser477C, Glu465C, Lys458C, Arg457C, Asn370C, Arg454C, Tyr421C, Tyr473C	
Saquinavir	ZINC29416466	Protease inhibitor	NDA #021785	NF	−10.7	Asp467C, Pro491C, Phe456C, Asn422C, Phe490C, Gln474C, Tyr473C, Lys458C, Glu465C, Try421C, Ile468C, Lys417C, Leu455C, Ala475C, Gly476C	
Suvorexant	ZINC49036447	Orexin receptor antagonist	NDA 204569	NF	−10.6	**Leu455C, Gln493****C**, Asn370C, Arg457C, Phe456C, Asp467C, Pro491C, Ile418C, Tyr421C, Tyr453C, Arg454C, Lys417C	
Riociguat	ZINC3819392	Guanylate cyclase stimulator	NDA 204819	NF	−10.6	**Leu455C, Gln493C**, Glu465C, Ile468C, Asp420C, Tyr473C, Tyr453C, Tyr421C, Arg457C, Asp467C, Arg454C, Phe456C, Lys417C	
Glibenclamide	ZINC537805	Insulin stimulator	NDA 017532	NF	−10.6	**Leu455C, Gln493C**, Tyr421C, Pro491C, Arg457C, Asp467C, Tyr473C, Glu465C, Arg454C, Phe456C, Tyr453C, Ile418C, Ile468C, Lys417C	
Candesartan	ZINC4074875	Angiotensin-II receptor antagonist	NDA #020838	NCT04351724 NCT04330300 NCT04394117 NCT04467931	−10.6	**Leu455C, Gln493C**, Phe456C, Arg457C, Glu465C, Lys458C, Asp467C, Ala475C, Ile468C, Phe490C, Gly476C, Tyr453C, Pro491C, Arg454C	

^a^ ID_ZINC (http://zinc.docking.org/), accessed March 2020. ^b^ Information retrieved from Drugs@FDA (https://www.accessdata.fda.gov), accessed September 2020. Latest New Drug Application (NDA) and Abbreviated New Drug Application (ANDA). ^c^ Information retrieved from https://clinicaltrials.gov/, accessed September 2020. ^d^ Bold letters indicate amino acids crucial for the formation of the spike glycoprotein-ACE2 complex.

**Table 4 molecules-25-05615-t004:** Selected drugs for the RBM in the up conformation.

Drug	ID_ZINC ^a^	Pharmacological Action	FDA-Approved Drugs ^b^	Clinical Studies Related to COVID-19 ^c^	Affinity (kcal/mol)	Amino Acid Interactions with Spike Glycoprotein ^d^	
Tedizolid Phosphate	ZINC43100953	Inhibition of bacterial protein synthesis	NDA #205435	NF	−9.7	Gln493B, Ser494B, Asp405B, ly416B, Ile402B, Ile418B, Tyr495B, Leu452B, Tyr451B, Tyr453B, Arg408B, Lys417B, Gln409B, Gln414B, Arg403B	
Atovaquone	ZINC116473771	Inhibitor of ubiquinol	ANDA #202960	NCT04339426 NCT04456153	−9.7	Leu455B, Ser494B, Ala352B, Asp467B, Phe456B, Asn450B, Pro491B, Thr478B, Gly476B, Ser477B, Tyr351B, Leu492B, Leu452B, Tyr449B, Arg454B, Pro479B	
Trimipramine	ZINC3831586	Selective serotonin reuptake inhibitors	ANDA #077361	NF	−9	Gln493B, Leu492B, Tyr449B, Val483B, Pro479B, Gly482B, Asn481B, Phe490B, Tyr489B, Phe490B, Cys488B	
Cefotetan	ZINC3830441	Inhibitors of bacterial wall synthesis	NDA #050588	NF	−8.8	Leu455B, Gln493B, Ser494B, Tyr489B, Pro491B, Gly476B, Ser477B, Phe490B, Leu492B, Tyr351B, Asp467B, Tyr449B, Asn450B, Arg454B, Thr478B, Pro479B, Leu452B, Phe456B, Tyr449B	
Losartan	ZINC3873160	Angiotensin II Type 1 Receptor Blocker (ARBs)	NDA #020386	16 studies found	−8.5	Arg355B, Glu516B, Asp428B, Asp427B, Phe429B, Tyr396B, Lys462B, Leu425B, Phe464B, Leu461B, Tyr423B, Asp398B, Pro426B, Pro463B, Phe429B	
Rosiglitazone	ZINC968330	Agonist of peroxisome proliferator-activated receptor	Discontinued	NF	−8.3	Gln493B, Ser494B, Asp405B, Ile402B, Tyr451B, Tyr453B, Ile418B, Gly416B, Thr415B, Gln414B, Arg408B, Gln409B, Glu406B, Tyr495B, Arg403B, Lys417B	

^a^ ID_ZINC (http://zinc.docking.org/), accessed March 2020. ^b^ Information retrieved from Drugs@FDA (https://www.accessdata.fda.gov), accessed June 2020. Latest New Drug Application (NDA) and Abbreviated New Drug Application (ANDA). ^c^ Information retrieved from https://clinicaltrials.gov/, accession June 2020. ^d^ Important amino acids involved for the formation of the spike glycoprotein-ACE2 complex.

**Table 5 molecules-25-05615-t005:** Selected drugs for the fusion site in the down conformation.

Drug	ID_ZINC ^a^	Pharmacological Action	FDA-Approved Drugs ^b^	Clinical Studies Related to COVID-19 ^c^	Affinity (kcal/mol)	Interactions with Spike Glycoprotein ^d^	
Saquinavir	ZINC29416466	Protease inhibitor	NDA #021785	NF	−11.1	**Cys851A, Leu849A**, Lys835A, Ala852A, Leu828A, Gly832A, Ile834A, Pro862A, Pro863A, Lys854A, Arg646C, Ile850A, Glu619C, Ser591C, Asp843A, Tyr837A, Asp614C, Asn616C, Val615C, Gln644C, Gly648C, Thr645C, Val860A, Tyr837A, NAG1309C	
Azilsartan	ZINC14210642	Angiotensin II Type 1 Receptor Blocker	NDA #200796	NCT04467931	−10.5	**Cys851A,****Leu849A**, Leu861A, Asp614C, Val860A, Lys835A, Ile834A, Thr866A, Ala831A, Ala829A, Arg646C, Asp867A, Thr866A, Gly832A, Ile850A, Pro863A, Pro862A, Ala668C, Ala852A, Leu828A, Tyr837A	
Loratadine	ZINC537931	H1-receptors inhibitor	NDA #021375	NF	−10.3	**Cys851A, Leu849A**, Asp614C, Gly832A, Pro862A, Pro863A, Thr866A, Ile850A, Arg646C, Asp867A, Ile834A, Phe833A, Lys835A, Leu828A, Val860A, Tyr843A, Ala668A	
Perampanel	ZINC30691797	Noncompetitive AMPA receptor antagonist	NDA #202834	NF	−10	**Cys851A, Leu849A**, Ile850A, Leu828A, Phe833A, Pro863A, Gly832A, Ala668A, Lys854A, Val860A, Arg646C, Tyr837A, Lys835A	
Aprepitant	ZINC27428713	Substance P/neurokinin 1 receptor antagonist	NDA #021549	NF	10	**Cys851A, Leu849A**, Ile834A, Pro862A, Ala668A, Leu861A, Gly832A, Ala831A, Ala829A, Thr866A, Asp843A, Asp614A, Leu849A, Cys851A, Val860A, Ala852A, Tyr837A, Ile850A, Tyr837A, Lys835A, Asp830A, Arg646C	
Nebivolol	ZINC4213946	β_1_-adrenergic receptor antagonist	NDA #021742	NCT04467931	−9.8	**Cys840B**, Asp848B, Ala845B, Gly842B, Gly1059B, Phe782B, Val729B, Thr778B, Pro1057B, Ser730B, Arg815B, Phe823B, Leu828B, Leu841B, Asp867B	
Terconazole *	ZINC3873936	Inhibiting de novo sterol biosynthesis	ANDA #075953	NF	−9.7	**Cys851A, Leu849A**, Asp614C, Pro862A, Pro863A, Thr866A, Ser813A, Gly836A, Ala852A, Gly832A, Ala831A, Glu868A, Arg646C, Asp867A, Leu849A, Ile850A, Val860A, Ala668C, Tyr837A, Lys854A, Lys835C, Ile834A	
Rolapitant	ZINC3816514	Substance P/neurokinin 1 receptor antagonist	NDA #206500	NF	−9.7	**Cys851A, Leu849A**, Ala647C, Leu861C, Ala668C, Pro863A, Ile834A, Gly832A, Tyr837A, Leu828A, Val869A, Asp614C, Arg646C, Arg646C, Pro862A, Ile850A, Ala852A, Pro862A, Val860A, Arg646C	
Difluprednate	ZINC4212945	Phospholipase A2 inhibitory	NDA #022212	NF	−8.9	**Cys840A, Cys851A, Leu849A**, Ala829A, Lys835A, Ala852A, Gly832A, Ile834A, Val860A, Asp614C, Ala668C, Thr866A, Ala647C, Phe833A, Arg646C, Ile850A, Leu828A, Tyr837A	

* Only cream and suppository vaginal presentation. ^a^ ID_ZINC (http://zinc.docking.org/), accessed March 2020. ^b^ Information retrieved from Drugs@FDA (https://www.accessdata.fda.gov), accessed September 2020. Latest New Drug Application (NDA) and Abbreviated New Drug Application (ANDA). ^c^ Information retrieved from https://clinicaltrials.gov/, accessed September 2020. ^d^ Bold letters indicate amino acids crucial for the FP function of the spike glycoprotein.

**Table 6 molecules-25-05615-t006:** Selected drugs for the fusion peptide in the up conformation.

Drug	ID_ZINC ^a^	Pharmacological Action	FDA-Approved Drugs ^b^	Clinical Studies Related to COVID-19 ^c^	Affinity (kcal/mol)	Amino Acid Interactions with Spike Glycoprotein ^d^	
Saquinavir	ZINC29416466	Protease inhibitor	NDA #021785	NF	−11	**Cys851C, Cys840C, Leu849C, Asp848C**, Asn856C, Asn960C, Gln853C, Leu858C, Ala852C, Thr859C, Asp614B, Thr732C, Phe833C, Val952C, Asn955C, Ile834C, Gly832C, Ala831C, Gln836C, Tyr837C, Arg847C, Lys854C, Val860C, Ala956C, Lys835C, Arg646B, Leu959C, Ala570B	
Maraviroc	ZINC100003902	Receptor CCR5 Antagonist	NDA #022128	NCT04435522 NCT04475991 NCT04441385	−10.7	**Cys851C, Cys840C, Leu849C, Asp848C**, Val952C, Asn955C, Leu959C, Thr732C, Leu858C, Ala570B, Asn856C, Val860C, Arg847C, Tyr837C, Gln836C, Arg646B, Lys835C, Ala956C, Phe833C, Val963C, Ala852C	
Azilsartan	ZINC14210642	ARBs	NDA 200796	NCT04467931	−10.6	**Cys851C, Cys840C, Leu849C****, Asp848C**, Tyr837C, Gln836C, Arg646B, Pro862C, Thr732C, Val952C, Asn955C, Phe833C, Arg847C, Thr859C, Lys854C, Leu849C, Val860C, Ala852C, Lys835C, Ala956C, Leu959C, Ile834C	
Tipranavir	ZINC100016058	Protease inhibitor	NDA #021814	NF	−10.5	**Cys851C, Cys840C, Leu849C,****Asp848C**, Thr734C, Ile834C, Lys835C, Thr732C, Phe855C, Asn856C, Gln853C, Arg646C, Gln836C, Tyr837C, Ala956C, Leu858C, Thr859C, Val963C, Asn960C, Val860C, Asp614C, Lys854C, Ala852C, Leu959C	
Ritonavir	ZINC3944422	Protease inhibitor	NDA #022417	89 studies found	−10.3	**Cys851C, Cys840C, Leu849C,****Asp848C**, Asn955C, Ile834C, Arg646C, Thr827C, Arg847C, Gly838C, Thr859C, Leu858C, Asp568B, Ile569B, Asn960C, Thr732C, Leu959C, Gln836C, Asp848C, Thr572B, Asn856C, Val952C, Ala956C, Ala852C, Ala570B, Val963C, Lys854C, Val860C, Lys835C, Tyr837C, Phe833C	
Isavuconazonium	ZINC29571072	Inhibit fungal cytochrome P450	NDA #207500	NF	−9.7	**Cys851C, Cys840C, Leu849C,****Asp848C**, Ala570B, Val963C, Leu959C, Val860C, Gly832C, Ala831C, Ile569B, Ser45C, Ser46C, Arg847C, Tyr837C, Gln836C, Lys835C, Asp57B, Asp586B, Gln853C, Ala852C, Thr572C, Asn856C, Lys557B, Asp568B, Asp568B, Phe833C, Ile834C	
Bosentan	ZINC1538857	Endothelin receptor antagonist	ANDA #205699	NCT04278404	−9.1	**Cys851C, Cys840C, Leu849C**, Asp848C, Thr827C, Ile834C, Gly832C, Arg847C, Arg646B, Val860C, Phe833C, Leu858C, Ile569B, Ala852C, Asp830C, Ala831C, Leu959C, Phe833C, Ala852C, Tyr837C, Gln836C	
Fosinopril sodium	ZINC3920355	angiotensin converting enzyme (ACE) inhibitor	NDA #019915 Discontinued	NCT04330300 NCT04467931	−9	**Cys851C, Cys840C, Leu849C**, Asp848C, Asn960C, Asn955C, Thr732C, Ile834C, Tyr837C, Lys835C, Asn960C, Val963C, Ala852C, Gln853C, Thr859C, Val860C, Gln836C; Lys854C, Arg646B, Ala956C, Leu959C, Leu858C, Phe833C, Arg847C	
Ceftazidime	ZINC3871960	β-lactamase inhibitors	ANDA #062655	NCT04278404	−8.7	**Cys851C, Cys840C, Leu849C**, Asp848C, Gly832C, Thr827C, Ile834C, Gln836C, Lys835C, Asn960C, Pro862C, Lys854C, Thr859C, Val860C, Arg847C, Arg646B, Tyr837C, Phe833C, Ala852C, Leu858C, Leu959C, Val963C	
Cefditoren pivoxil	ZINC4215234	Cell wall inhibitor	NDA 021222	NF	−8.4	**Cys851C, Cys840C, Leu849C**, Asp848C, Leu828C, Asp830C, Thr827C, Ala829C, Ile834C, Asp848C, Lys835C, Phe833C, Gly838C, Val860C, Tyr837C, Arg847C, Arg646B, Gln836C, Gly832C, Ala853C	

^a^ ID_ZINC (http://zinc.docking.org/), accessed March 2020. ^b^ Information retrieved from Drugs@FDA (https://www.accessdata.fda.gov), accessed September 2020. Latest New Drug Application (NDA) and Abbreviated New Drug Application (ANDA). ^c^ Information retrieved from https://clinicaltrials.gov/, accessed September 2020. ^d^ Bold letters indicate amino acids crucial for the FP function of the spike glycoprotein.

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
