# Peer review of "Repositioning of Ligands That Target the Spike Glycoprotein as Potential Drugs for SARS-CoV-2 in an In Silico Study"

_molecules, 2020, doi:10.3390/molecules25235615_

Round 1

Reviewer 1 Report

The manuscript molecules-997827, Repositioning of ligands that target spike glycoprotein as potential drugs against SARS-CoV-2 could be interesting for the drug developing efforts towards COVID19, but it lacks the proper scientific rigor to be published. The authors seems to lack the pharmacology expertise because there are many mistakes in that field, especially in the “Discussion” section. I pointed out just some of the mistakes, the authors should carefully read their paper and maybe use the help of a medicinal chemistry or pharmacology specialist.

I don’t think the authors chose the best title. The title should clearly indicate that this is a simulation towards the spike glycoprotein. There are not experimental data to prove that these compounds actually target the aforementioned protein.

The introduction should be extended. There are some important articles that should be referenced by the authors, works that could help the readers better understand the therapeutically solutions for COVID-19. Here are some examples that could be useful:

SARS-CoV-2 pathophysiology and its clinical implications: An integrative overview of the pharmacotherapeutic management of COVID-19, Food and Chemical Toxicology, Volume 146, December 2020, 111769

Comprehensive analysis of drugs to treat SARS‑CoV‑2 infection: Mechanistic insights into current COVID‑19 therapies (Review), Int J Mol Med 2020, https://doi.org/10.3892/ijmm.2020.4608

Check the references. Some articles appear twice. See for example Novac (2013).

In table 3, and elsewhere in the paper, change “diabeta” to the noncommercial drug name of glibenclamide. The same for “accolade” that should be changed to zafirlukast. The authors should not use any commercial names. Please check the manuscript!

Row 32, Sulbactam is not really an antibiotic. It is used with antibiotics that are inactivated by bacteria using β-lactamase Please verify!

Row 89, “we repositioned drugs” is not a proper use of the concept. The authors propose some new uses for some drugs, but they are not approved for that use. The authors should be careful not to suggest the therapeutically use of any of these drugs for COVID-19 patients!

The format of table 1 and other tables is not that of the journal. The authors should check the editing of each table according to the journal format and correct them. In table 1 the pharmacological use of varenicline is not correctly described. It should be a “α4β2 nicotinic e receptor agonist”.

In figure 4A, the structure of Tigecycline looks strange. Please check it. Also it is an important problem that the authors seems to ignore. There are multiple possible Keto-enol tautomers. Were they accounted for? There are possible several intern hydrogen bonds. Were the intramolecular hydrogen bonds considered when docking? The authors should answer to this question for all the tested compounds. For example, glibenclamide. This problem could lead to serious errors!

In Table 3, Candesartan is presented as “ACEII antagonist”, but correct should be “angiotensin II receptor antagonist”. Similarly in Table 4, Losartan is described wrongly as ACE antagonist and in Table 5 Azilsartan is presented as ACE2 Blocker. In the abbreviations, ACE2 is presented as angiotensin-converting enzyme 2, not to be confused with angiotensin II. The confusion seems to be present also in other sections of the paper. See row 185.

The discussion section is overdeveloped by presenting confusing information. Some of the drugs presented in these study are under clinicaltrials targeting other mechanism than the spike glycoprotein. The authors present these data as a conformation of their study. It is not! And it can be confusing. I would advise the authors to remove all discussion on 3C-like protease or similar.

On row 68, “In another study pancuronium is included as an anesthetic in pregnant and postpartum women hospitalized with flu syndrome. Therefore, the present study reports an additional mechanism of these drugs to avoid SARS-CoV-2 infection”. I consider this statement wrong. Firstly, it has no reference to check its content. Secondly Pancuronium (it should be pancuronium bromide) is not used as anesthetic! It is a muscle relaxant! Thirdly, the use in flu patients does not mean it has any effect on SARS-CoV-2.

Another example “Atovaquone is in a clinical trial (NCT04339426, in combination with Azithromycin) to treat COVID-19, due to its ability to treat or prevent pneumonia caused by Pneumocystis carinii”. This is a clear logical fallacy. If Atovaquone is use for a special type of pneumonia, it can be used for any type of pneumonia?

Reviewer 2 Report

The authors of manuscript entitled “Repositioning of ligands that target spike glycoprotein as potential drugs against SARS-CoV-2” conducted an analysis of drugs in search of their other therapeutic applications, for the treatment of COVID-19 disease.

In the submitted article, they described an in silico strategy called "drug repositioning", in which they searched for drugs that were available and could target spike glycoprotein. Taking into account the widely discussed literature and the presented research results, a conclusion can be drawn about the possibility of using multi-target drugs, because interference at different points in the replication cycle may bring a better effect. The obtained information can help to plan and optimize the treatment strategy (combination therapies), as well as indicate the direction of work on the vaccine, taking into account the structural conditions of combining drugs with appropriate receptors.

Presented manuscript may be accepted for publication after minor corrections.

Comments:

p.1 - Why are the names of some active substances written with a capital letter, and others with a lower case?

p.18 line 95 – please correct: “…fusion function [24](Lai et al, 2017). In addition, these…”

References - There are entries 32 through 64 in the reference list that are not cited in the manuscript. Are they here accidentally, or supposed to be included in the work?

Round 2

Reviewer 1 Report

The authors made important changes to the paper according to the comments of the reviewers. Still the problem of multiple possible keto-enol tautomers or the possibility of several intern hydrogen bonds was not adressed. Were these intramolecular hydrogen bonds considered when docking? The authors should discuss in their manuscript this problem for the readers to understand this weak point.

The style of the manuscript needs to be edited to that of the Molecules Journal

Author Response

Point 1: The authors made important changes to the paper according to the comments of the reviewers. Still the problem of multiple possible keto-enol tautomers or the possibility of several intern hydrogen bonds was not adressed. Were these intramolecular hydrogen bonds considered when docking? The authors should discuss in their manuscript this problem for the readers to understand this weak point.

Response 1, Thank you for the comments.  To this work we used the drugs located at (http://zinc.docking.org) which has different formats (https://www.ncbi.nlm.nih.gov/pmc/articles/PMC1360656/), for our docking studies we employed the 3D data. All small molecules have an ID which is included into the Tables which list the corresponding promissory drugs. Due to our objective was to assays in silico those small molecules placed at ZINC database on Spyke protein of SARS-Cov-2, we do not consider their tautomers, stereoisomers, enantiomers, different conformational motions etc.  Regarding to hydrogen bonds, Autodock is a program used for docking studies of small molecules on macromolecules that has force field (for scoring function) which consider non bond (atom-atom bonds, angles etc.) and non-bond interactions (hydrogen bond, hydrophobic interactions, solvatation, electrostatic interactions, etc.) to measure the binding free energy (DG) values (http://autodock.scripps.edu/faqs-help/faq/where-do-i-set-the-autodock-4-force-field-parameters) which was reported elsewhere (Morris, G. M., Goodsell, D. S., Halliday, R.S., Huey, R., Hart, W. E., Belew, R. K. and Olson, A. J. Automated Docking Using a Lamarckian Genetic Algorithm and and Empirical Binding Free Energy Function. J. Comput. Chem., 1998, 19, 1639-1662. 2. Huey, R., Morris, G. M., Olson, A. J. and Goodsell, D. S. A Semiempirical Free Energy Force Field with Charge-Based Desolvation. J. Comput. Chem., 2007, 28, 1145-1652. 3. Morris, G.M., Huey, R., Lindstrom, W., Sanner, M.F., Belew, R.K., Goodsell, D.S. and Olson, A.J. AutoDock4 and AutoDockTools4: Automated docking with selective receptor flexibility. J. Comput. Chem. 2009, 30, 2785-2791). However, for better understanding of the ligand-protein complex, we have described the non bond interactions considering those < than 3 Angstroms.

In this latest version, the text of the article was not modified, only the reply letter to the reviewer was added.